# A dynamic interaction between CD19 and the tetraspanin CD81 controls B cell co-receptor trafficking

Katherine J Susa[1], Tom CM Seegar[1], Stephen C Blacklow[1,2]*, Andrew C Kruse[1]*

[1]Department of Biological Chemistry and Molecular Pharmacology, Blavatnik Institute, Harvard Medical School, Boston, United States; [2]Dana Farber Cancer Institute, Department of Cancer Biology, Boston, United States

**Abstract** CD81 and its binding partner CD19 are core subunits of the B cell co-receptor complex. While CD19 belongs to the extensively studied Ig superfamily, CD81 belongs to a poorly understood family of four-pass transmembrane proteins called tetraspanins. Tetraspanins play important physiological roles by controlling protein trafficking and other processes. Here, we show that CD81 relies on its ectodomain to traffic CD19 to the cell surface. Moreover, the anti-CD81 antibody 5A6, which binds selectively to activated B cells, recognizes a conformational epitope on CD81 that is masked when CD81 is bound to CD19. Mutations of CD81 in this interface suppress its CD19 export activity. These data indicate that the CD81 - CD19 interaction is dynamically regulated upon B cell activation and this dynamism can be exploited to regulate B cell function. These results are not only valuable for understanding B cell biology, but also have important implications for understanding tetraspanin function generally.

*For correspondence:
stephen_blacklow@hms.harvard.edu (SCB);
Andrew_Kruse@hms.harvard.edu (ACK)

## Introduction

The tetraspanins constitute a 33-member family of transmembrane proteins in humans. Although poorly understood, tetraspanins play a critical role in mammalian physiology, functioning in nearly all cell types and regulating distinct processes such as control of cell morphology, cell adhesion, protein trafficking, and signal transduction (*Hemler, 2008*). Tetraspanins are thought to achieve their biological functions through interactions with partner proteins, leading to formation of signaling complexes and modulation of signaling activity (*Hemler, 2005*). Members of the tetraspanin protein family share an overall domain organization consisting of four transmembrane segments, a small extracellular loop (SEL), a large extracellular loop (LEL) containing a conserved Cys-Cys-Gly (CCG) motif, a short cytoplasmic N-terminal region, and a C-terminal cytoplasmic tail. However, the molecular details of how these domains mediate complex formation with partner proteins to regulate their trafficking and signaling remain unclear.

CD81, the first tetraspanin identified, was discovered as the target of an antiproliferative antibody called '5A6', which inhibits the growth of B cell lymphoma cell lines (*Oren et al., 1990*). CD81 plays a critical role in regulating B cell receptor (BCR) signaling as one subunit of the B cell co-receptor complex, which also includes CD19 and CD21 (*Carter and Barrington, 2004*). Within this complex, CD81 directly interacts with CD19, a single-pass transmembrane protein that establishes the threshold for both BCR dependent and independent signaling. Stimulation of CD19 lowers the signaling threshold needed for both antigen-independent and antigen-dependent activation of B cells by several orders of magnitude, and this signaling is critical for the function of the humoral immune response (*Gauld et al., 2002*; *Carter and Fearon, 1992*). Not surprisingly, aberrant CD19 signaling is implicated the development of B cell malignancies, autoimmunity, and immunodeficiency (*Barrena et al., 2005*; *Yazawa et al., 2005*; *Mei et al., 2012*; *van Zelm et al., 2006*). CD19 is also

the target of chimeric antigen receptor expressing T cells now used clinically in the treatment of B cell malignancies (*Brentjens et al., 2013*; *Grupp et al., 2013*; *Kalos et al., 2011*; *Kochenderfer et al., 2012*; *Porter et al., 2011*).

Despite the importance and therapeutic relevance of the B cell co-receptor, surprisingly little is known about how CD81 engages CD19 to regulate its trafficking or signaling activity. The association between CD19 and CD81 was first detected using co-immunoprecipiation studies (*Bradbury et al., 1992*). Later, genetic evidence revealed that defects in complex formation between CD19 and CD81 result in severe deficiencies in B cell function. For example, three independent lines of CD81-null mice showed reduced CD19 surface expression accompanied by defects in B cell function such as weaker early antibody responses, impaired B cell proliferation, and reduced calcium influx following B cell activation (*Maecker and Levy, 1997*; *Tsitsikov et al., 1997*; *Miyazaki et al., 1997*). Additionally, there are human cases of common variable immune deficiency (CVID) in which CD19 expression on B cells is suppressed by homozygous truncations in the CD81 gene (*van Zelm et al., 2010*).

It has been proposed that CD81 has two key roles as a B cell co-receptor subunit. First, it is thought to chaperone CD19 through the secretory pathway to the plasma membrane (*Braig, 2016*; *Shoham et al., 2003*). Second, it may also serve as a regulator of B cell signaling by controlling the localization of CD19 at the plasma membrane during B cell activation (*Mattila et al., 2013*). The molecular mechanisms by which CD81 carries out both trafficking of CD19 and regulation of its signaling activity, however, remain unclear.

Here, we find that CD81 uses its ectodomain to bind CD19 and to promote the export of CD19 to the cell surface. Remarkably, the anti-CD81 antibody 5A6, which binds selectively to activated B cells, recognizes an unusual conformational epitope on CD81 that is masked when CD81 is in complex with CD19, but which becomes accessible upon B cell activation. These findings suggest that the CD81 - CD19 interaction is dynamic and is linked to the B cell activation state.

## Results

### The large extracellular loop of CD81 is required to promote CD19 export to the cell surface

Prior studies have suggested that the first TM helix of CD81 is the main specificity determinant for trafficking of CD19 to the cell surface (*Shoham et al., 2006*), but this claim has also been called into question (*Berditchevski and Odintsova, 2007*). To determine which regions of CD81 are necessary for trafficking CD19 to the cell surface, we established a HEK293T cell line in which CD81 was knocked out with CRISPR-Cas9 and used flow-cytometry to test the ability of CD81 or CD81 chimeric proteins to enhance delivery of CD19 to the cell surface. We created chimeras of CD81 with CD9 (the tetraspanin most similar to CD81 in sequence) and with Tspan15 (a divergent tetraspanin from *C. elegans*). The chimeras have domain swaps of the small extracellular loop, large extracellular loop, or first transmembrane helix (*Figure 1A*). Expression of CD81 chimeras was confirmed by flow cytometry (*Figure 1—figure supplement 1*).

Whereas cells transfected with CD19 alone only show a small amount of surface staining (*Figure 1B and C*), co-transfection of CD19 with wild-type CD81 results in a two to four-fold enhancement in CD19 surface staining. Chimeras that retain the large extracellular loop of CD81 stimulate the same increase in CD19 surface staining as wild type CD81, but chimeras lacking the large extracellular loop of CD81 do not, indicating that the large extracellular loop is necessary for CD19 surface export. Replacement of the first TM helix of CD81 with that of CD9 or Tspan15 also supports the same increase in surface staining as wild type CD81, whereas replacement of the first TM helix of CD9 with that of CD81 fails to increase surface staining of CD19, revealing that the first TM helix of CD81 in the context of a CD9 backbone is not sufficient to support the trafficking of CD19 (*Figure 1B and C*).

To assess whether surface export of CD19 depends upon the presence of its native TM region, we created a CD19 chimera in which only its native TM (residues 292–313) was replaced with that of the σ1 receptor (residues 6–32) (*Schmidt et al., 2016a*). CD81 exports the CD19/σ1 chimera to the cell surface as effectively as it exports wild-type CD19, indicating that it is association of the ectodomains that promotes the trafficking of CD19 to the cell surface (*Figure 1D*). A secreted form of the

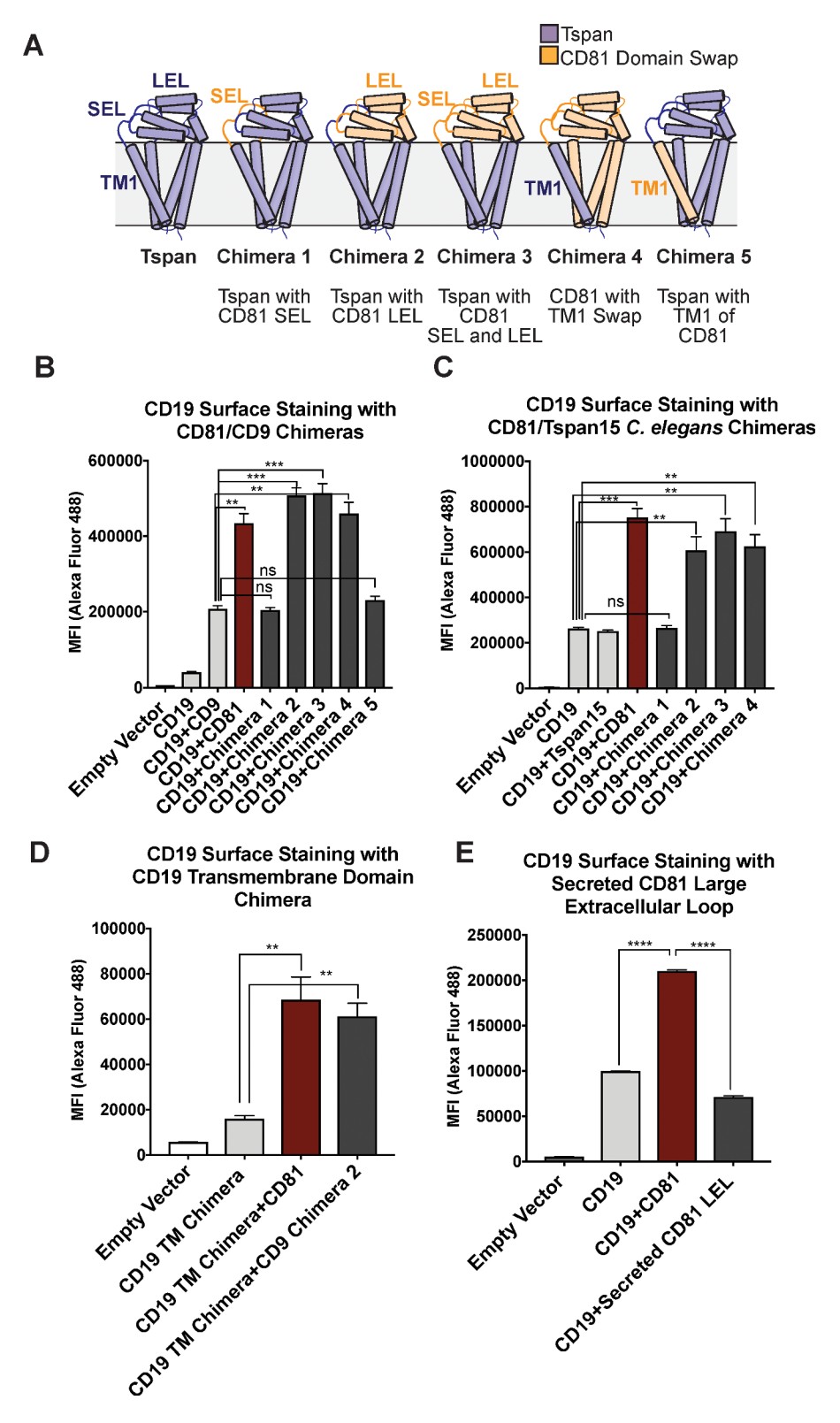

**Figure 1.** CD81 chimera design and CD19 Export Assay. (**A**) Design of CD81 Chimeras used in export assay experiments. (**B**) Export assay with CD81/CD9 chimeras. (**C**) Export assay with CD81/Tspan15 *C. elegans* chimeras. (**D**) Export assay with CD19/ σ1 receptor transmembrane domain chimera. (**E**) Export assay with a secreted construct of the CD81 large extracellular loop. For the data in panels (**B** – **E**), surface CD19 was detected by flow cytometry

*Figure 1 continued on next page*

*Figure 1 continued*

using an Alexa 488-coupled anti-CD19 antibody. Each figure represents three independent experiments. Error bars represent mean ± SEM. Statistical analysis was performed in GraphPad Prism using an unpaired two-tailed t test. **p<0.01; ***p<0.001, ****p<0.0001.

The online version of this article includes the following figure supplement(s) for figure 1:

**Figure supplement 1.** Surface staining of CD81 chimeras used in the CD19 Export Assay.

**Figure supplement 2.** Representative gating strategy for CD81 null 293 T cells used in the CD19 export assay and CD19-CD81 fusion protein validation experiments.

CD81 large extracellular loop, however, does not promote increased CD19 surface expression, suggesting that membrane tethering plays an important role in CD19 surface delivery by increasing the effective concentrations of the two proteins for each other (*Figure 1E*).

There are several possible explanations for why our findings differ from those in a previous report claiming that the first TM helix of CD81 is the main specificity determinant for trafficking of CD19 to the cell surface (*Shoham et al., 2006*). First, our chimeras were designed using the domain boundaries informed by the crystal structure of full-length CD81, which was not available at the time of the previous work. Second, the prior studies only reported chimeras between CD81 and CD9, which is the tetraspanin most similar in sequence to CD81, whereas we have made structurally informed chimeras with even more distantly related tetraspanins.

## The epitope of the 5A6 CD81 antibody is masked when CD81 is in complex with CD19

To further characterize the CD19-CD81 complex, we constructed a fusion protein in which the C-terminus of CD19 is directly connected to the N-terminus of CD81 with a short intervening linker (*Figure 2A*). To assess the integrity of this fusion protein, we evaluated its abundance on the cell surface by flow cytometry, and examined the reactivity of the fusion protein with a panel of anti-CD19 and anti-CD81 antibodies (*Nelson et al., 2018*). The abundance of the CD19-CD81 fusion protein on the cell surface is comparable to that observed when full-length CD19 is co-expressed with wild type CD81, indicating that CD81 in the fusion protein is functional in trafficking CD19 to the cell surface (*Figure 2B*). Moreover, four different anti-CD19 antibodies recognize the fusion protein (*Figure 2—figure supplement 1*), as do three anti-CD81 antibodies, providing further evidence that both CD19 and CD81 are properly folded in the context of the fusion protein. Surprisingly, however, one anti-CD81 antibody, called 5A6 (*Levy, 2017*; *Oren et al., 1990*), showed significantly decreased binding of the fusion protein compared to the other anti-CD81 antibodies (*Figure 2C*). Although all four antibodies bind the large extracellular loop of CD81, only 5A6 is unable to detect the CD19-CD81 fusion protein, suggesting that its epitope overlaps with the region(s) of CD81 that contact CD19 in the native complex (*Nelson et al., 2018*). A prior co-immunoprecipitation experiment also showed that the 5A6 antibody cannot be used to pull down the components of CD21/CD19 complex in a B cell line, providing further evidence the 5A6 epitope is masked by CD19 (*Matsumoto et al., 1993*).

## 5A6 binds to helices C and D of CD81

To gain insight into the molecular basis underlying the unique reactivity of the 5A6 antibody, we determined the structure of the 5A6 $F_{ab}$ in complex with the large extracellular loop (LEL) of CD81 to 2.4 Å resolution using x-ray crystallography (*Table 1*). The overall architecture of the CD81 LEL has five helices, with the A, B and E helices as a stalk and helices C and D capping the 'top' face. The 5A6 $F_{ab}$ binds CD81 at an epitope derived almost exclusively from helices C and D (*Figure 3A, B*), burying a total of 1522 Å (*Hemler, 2005*) of solvent accessible surface area. The paratope of 5A6 is derived from all three heavy chain complementarity-determining regions (CDRs; residues 31–35, 50–66, 99–108) and from the first two light light-chain CDRs (residues 24–40 and 55–61).

The most striking feature of the contact interface is the large-scale rearrangement of the CD81 C and D helices, which splay apart in the structure of the complex (*Figure 3D*). Helix C moves outward by approximately 7 Å, and Helix D unravels almost completely and moves outward by approximately 11 Å, compared to its position in the structure of free full-length CD81. This structural rearrangement occurs because the heavy chain of 5A6 inserts its CDR3 loop between the helices, allowing it

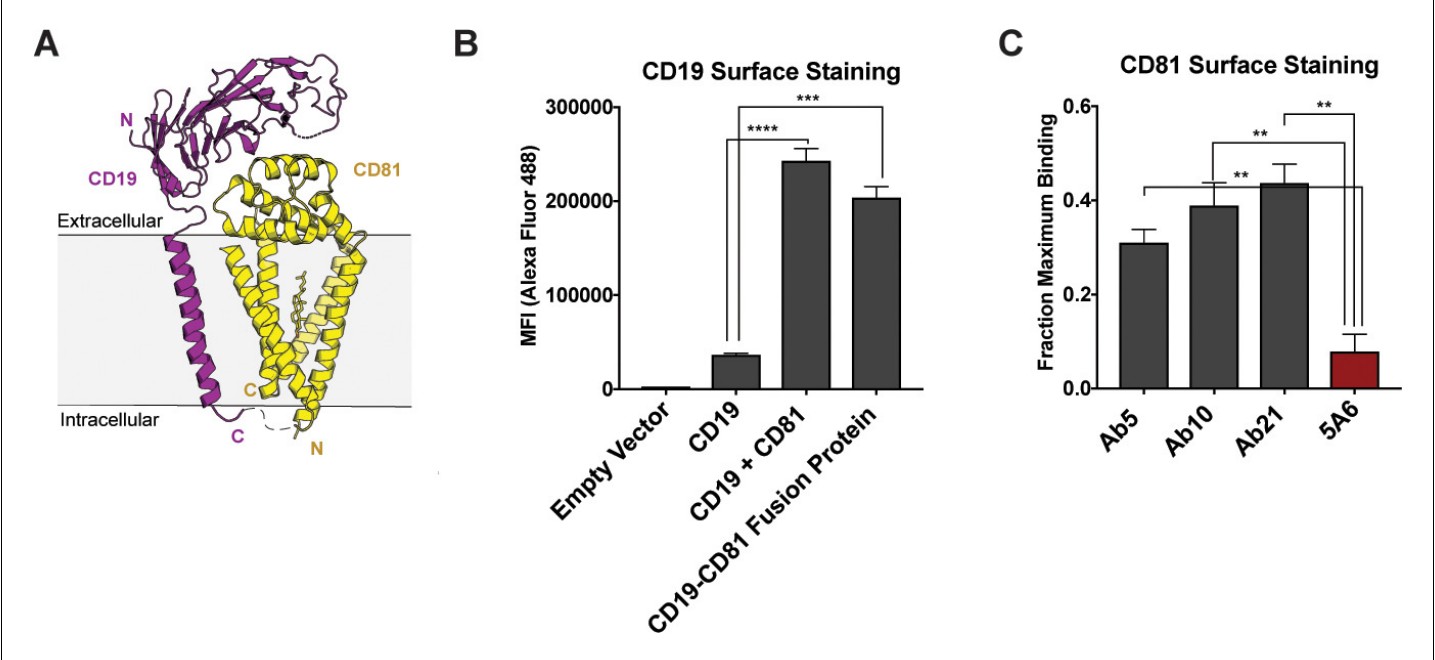

**Figure 2.** Design and evaluation of a CD19-CD81 fusion protein. (**A**) Cartoon representing the designed CD19-CD81 fusion protein. The model was created based on known structures of the CD19 ectodomain (PDB 6AL5) and CD81 (PDB 5TCX). A short Gly-Ser linker (dashed lines) connects P329 of the intracellular portion of CD19 to the N-terminus of CD81. (**B**) Analysis of CD19 surface staining in CD81-null cells expressing the CD19-CD81 fusion protein. Surface staining for the CD19-CD81 fusion is compared to staining of cells expressing only CD19, and to staining of cells expressing both CD19 and CD81, using an Alexa 488-coupled anti-CD19 antibody. (**C**) Binding of various CD81 antibodies to the CD19-CD81 complex, analyzed by flow cytometry. 'Fraction maximum binding' was calculated by dividing the average MFI of antibody bound to CD19-CD81 by the average MFI of antibody bound to CD81. An anti-human IgG-Alexa 488 secondary antibody was used to detect CD81 antibody bound to the cell surface. For the data in panel B and C, each figure represents three independent experiments and error bars represent mean ± SEM. Statistical analysis was performed in GraphPad Prism using an unpaired t test. **p<0.01; ***p<0.001, ****p<0.0001.

The online version of this article includes the following figure supplement(s) for figure 2:

**Figure supplement 1.** Validation of the CD19-CD81 fusion protein with a panel of CD19 antibodies.

to form polar contacts with S179 and N180. Additional key interactions at the $F_{ab}$-CD81 interface include extensive light-chain contacts with helix C of CD81 (*Figure 3C*). Among these interactions are hydrogen-bonds between the side chain hydroxyl group of $F_{ab}$ residue Y55 with the T167 side chain hydroxyl, the T167 backbone amide, and the T163 backbone carbonyl of CD81. Side chain hydrogen bonding interactions are also present between T59 of the $F_{ab}$ and T163 of CD81, and between S62 of the $F_{ab}$ and S168 of CD81. The light chain of 5A6 also contacts three residues at the start of Helix E, forming a hydrogen bonding network with residues E188, D189, and Q192 of CD81.

## Helix C and D of CD81 mediate CD19 complex formation

The binding of 5A6 to CD81 results in substantial conformational changes of Helices C and D in CD81. These two helices form a solvent-exposed, low polarity region in the CD81 large extracellular loop, and evolutionary analysis reveals that this region is highly variable among the different proteins of the tetraspanin family (*Figure 3B*). Both molecular dynamics simulations and NMR studies suggest the Helix D of CD81 is the most flexible region of the large extracellular loop (*Rajesh et al., 2012*; *Schmidt et al., 2016b*). In molecular dynamics simulations, Helices A, B, and E from the large extra-cellular loop of CD81 retain their alpha helical structure, but Helices C and D are more labile and show a tendency to lose alpha-helicity (*Schmidt et al., 2016b*).

Because the 5A6 antibody is non-reactive with the CD19-CD81 fusion protein, and because its epitope consists primarily of the CD81 C and D helices, we hypothesized that the C/D helix region of CD81 is essential for CD19 binding. To test this idea, we introduced alanine substitutions in helix C or D and measured the effect of these mutations on export of CD19 to the cell surface using our

**Table 1.** X-ray crystallography data collection and refinement statistics.
Refined coordinates and structure factors are deposited in the Protein Data Bank under accession code 6U9S.

| Data collection | 5A6-CD81 LEL |
| --- | --- |
| Wavelength (Å) | 0.9792 |
| Space Group | P $2_1$ $2_1$ $2_1$ |
| Number of crystals | 1 |
| Unit cell dimensions | |
| a,b,c | 40.003, 96.858, 297.091 |
| α, β, γ (°) | 90, 90, 90 |
| Resolution (Å) (last shell) | 49.5–2.4 (2.54–2.4) |
| No. of reflections (total/unique) | 293920/46497 |
| Completeness (%) (last shell) | 99.1 (96.8) |
| I/σ(I) (last shell) | 7.94 (0.46) |
| Rmeas (%) (last shell) | 20.5% (355.7%) |
| $CC_{1/2}$ (%) (last shell) | 99.5 (16.6) |
| Multiplicity | 6.3 |
| Refinement | |
| Number of atoms (protein/solvent) | 7974/358 |
| Rwork/Rfree (%) | 21.44/28.08 |
| R.M.S. deviation (Å) | |
| Bond length | 0.003 |
| Bond angles | 0.537 |
| Ramachandran statistics | |
| Favored | 96.94 |
| Allowed | 3.06 |
| Outliers | 0.00 |

flow cytometry assay. Mutation of either helix C or D to polyalanine results in decreased trafficking of CD19 to the cell surface, strongly suggesting that each helix contributes to CD19-CD81 complex formation (*Figure 3E*).

## The CD19-CD81 complex dissociates in activated B cells

Although CD81 has a clear role in trafficking CD19 to the cell surface, its function at the B cell membrane remains unclear. Tetraspanins are thought to organize receptors and associated signaling proteins in functional microdomains in the plasma membrane, thereby regulating receptor signaling and their associated signaling pathways. Super-resolution microscopy suggests that CD19 may be compartmentalized in the B cell membrane by CD81 to regulate signaling through the BCR, but the mechanistic details of how CD81 regulates the localization of CD19 remain poorly understood (*Mattila et al., 2013*; *Zuidscherwoude et al., 2015*).

To address this question, we used antibodies with different CD81 binding epitopes to probe the dynamics of the CD19-CD81 complex on primary human B cells in response to B cell activation. We isolated primary human B cells from a fresh leuko-reduction collar and activated them with an anti-B cell receptor (BCR) antibody. Anti-BCR antibody treatment resulted in increased CD69 and CD86 at the cell surface when compared with resting cells, confirming activation of the antibody-stimulated cells (*Figure 4A*). We then compared the surface staining of CD19 and CD81 in the resting and activated states, using anti-CD81 antibodies with different epitopes to distinguish free CD81 from CD19-associated CD81. CD19 showed no difference in surface staining between resting and activated cells (*Figure 4B*). When surface CD81 is detected using Ab21 (*Nelson et al., 2018*), which

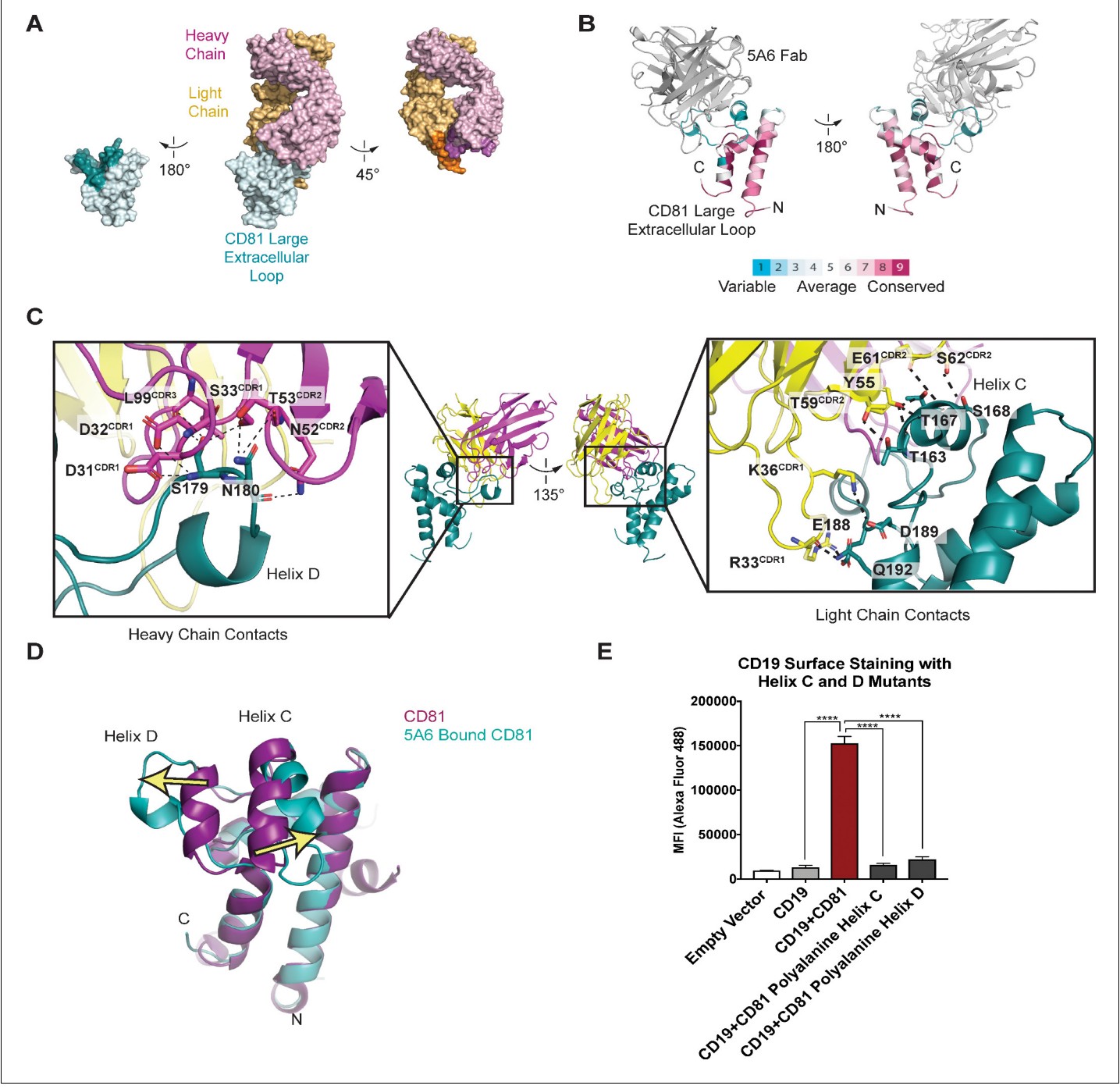

**Figure 3.** Structure of the 5A6 $F_{ab}$-CD81 Large Extracellular Loop Complex (PDB 6U9S). (**A**) Surface representation of the 5A6-CD81 complex. CD81 is blue, the 5A6 $F_{ab}$ light chain is yellow, and the heavy chain is magenta. Residues at the binding interface are colored in a darker shade. (**B**) CD81 colored by evolutionary conservation score using the top 50 CD81-related sequences determined by Consurf (*Landau et al., 2005*).(**C**) 5A6 $F_{ab}$-CD81 binding interface. Heavy chain (left panel) and light chain (right panel) contacts are shown. Hydrogen bonding interactions are indicated with dotted lines. (**D**) Structural superposition of 5A6 bound CD81 on full length CD81. Arrows indicate positional shifts of the variable helices C and D in the 5A6-bound structure. (**E**) CD19 Export Assay with Helix C and D Mutants. Surface CD19 was detected by flow cytometry using an Alexa 488-coupled anti-CD19 antibody. Expression of helix C and D mutants was confirmed by flow cytometry (*Figure 3—figure supplement 3*). For the data in panel E, error bars represent mean ± SEM of three independent experiments. Statistical analysis was performed in GraphPad Prism using an unpaired t test. \*\*p<0.01; \*\*\*p<0.001, \*\*\*\*p<0.0001.

The online version of this article includes the following figure supplement(s) for figure 3:

**Figure supplement 1.** Representative Density in the CDRs of 5A6 $F_{ab}$.

*Figure 3 continued on next page*

*Figure 3 continued*

**Figure supplement 2.** Epitope comparison of Ab5, Ab10, Ab21, and 5A6.
**Figure supplement 3.** CD81 surface staining of polyalanine mutants detected with Ab21.

recognizes both free and complexed CD81, there is no difference between resting and activated B cells (*Figure 4B and C*), but when CD81 is detected using the 5A6 antibody, which selectively recognizes free CD81, there is a two-fold increase in surface staining on activated B cells (*Figure 4B and C*). Western blotting of whole cell lysates from resting and activated cells with Ab21 and 5A6 revealed no significant difference in protein levels, indicating that the increase in CD81 surface

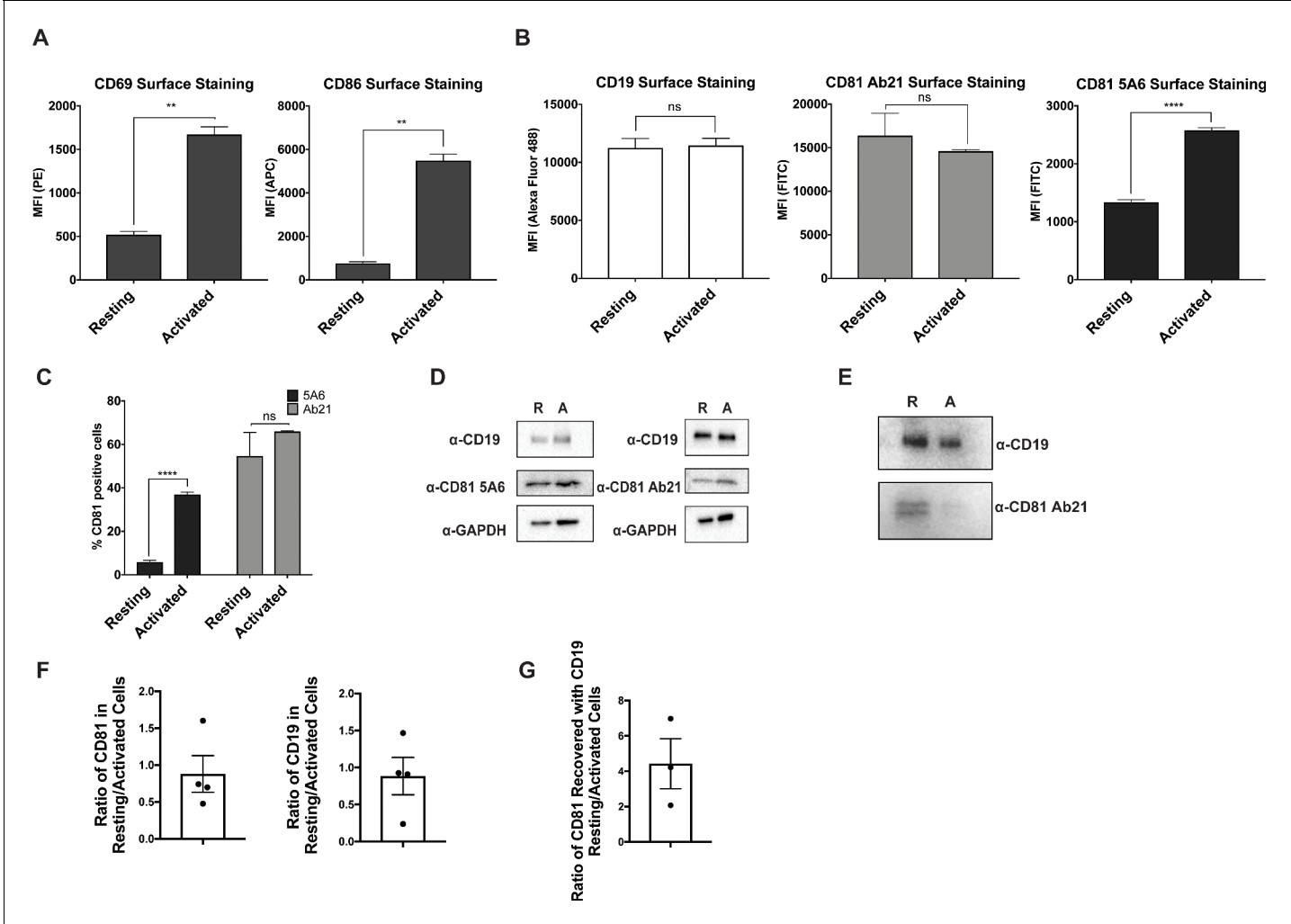

**Figure 4.** CD81 antibody labeling experiments in resting and activated primary human B cells. (A) Surface staining of the B cell activation markers, CD86 and CD69, in resting B cells and cells activated with IgM, IgG Fab'2. (B) Surface staining of CD19 and CD81 with antibody 5A6 and Ab21. (C) Percent of CD81 positive cells labeled with 5A6 or Ab21. (D) Western blots of total protein lysate. 'R' represents resting cells and 'A' represents activated cells. (E) Immunopurification of CD19 from resting and activated primary human B cells, followed by western blotting for CD19, and for CD81 using Ab21. 'R' represents resting cells and 'A' represents activated cells. (F) Densitometry analysis of western blots of whole cell lysates shown in Panel D. (G) Densitometry analysis of western blots of CD19-CD81 co-immunoprecipitation shown in Panel E. For all panels, data are shown as mean ± SEM. Three replicates were performed for CD81 5A6 and Ab21 staining, and two replicates were performed for CD19, CD69, and CD86 staining conditions. Statistical analysis was performed in GraphPad Prism using an unpaired t test. **p<0.01; ***p<0.001, ****p<0.0001.
The online version of this article includes the following figure supplement(s) for figure 4:

**Figure supplement 1.** Replicate western blots.
**Figure supplement 2.** Representative gating strategy for primary human B cells.

staining is not due to increased production of CD81 in activated B cells (*Figure 4D*) but instead reflects a change in the accessibility of the 5A6 epitope. This finding indicates that CD81 either undergoes a conformational change or dissociates from CD19 to expose the epitope upon B cell activation. To distinguish between these two possibilities, we immunopurified CD19 from resting and activated primary B cells. Immunoprecipitation of CD19 recovered much more CD81 from resting B cells than it did from activated B cells, indicating that the CD19-CD81 complex dissociates in activated B cells (*Figure 4E*).

## Discussion

Tetraspanins control a wide range of physiological processes by interacting with partner proteins (*Hemler, 2005*), yet there is remarkably little structural or mechanistic information about how tetraspanins bind and regulate their molecular partners. Here, using a combination of molecular engineering, X-ray crystallography, and cell-based assays we investigated the interaction between the prototypical tetraspanin CD81 and its biochemical partner CD19, the key signaling subunit of the B cell co-receptor complex.

Our studies show that the ectodomains of CD19 and CD81 interact dynamically during B cell co-receptor trafficking and signaling upon B cell activation. Our structure of the complex between the 5A6 F$_{ab}$ and the extracellular domain of CD81 reveals that 5A6 binds to an unusual conformational epitope, which is masked in the CD81-CD19 complex. Other CD81 antibodies have epitopes in nearby regions of the CD81 large extracellular loop, yet none rely exclusively on helix C and D for binding or approach CD81 from a similar angle (*Figure 3—figure supplement 2*), suggesting that the exact details of antigen recognition geometry give rise to the unique properties of 5A6. Using immunoprecipitation and flow cytometry, we found that the association between CD19 and CD81 is dynamic and that CD19 dissociates from CD81 after B cell activation (*Figure 5*). This information could be exploited to develop novel co-receptor antibody therapeutics to selectively target activated B cells and to guide development of conformationally selective antibodies targeting other therapeutically relevant tetraspanin-partner protein complexes. Indeed, the 5A6 antibody itself has

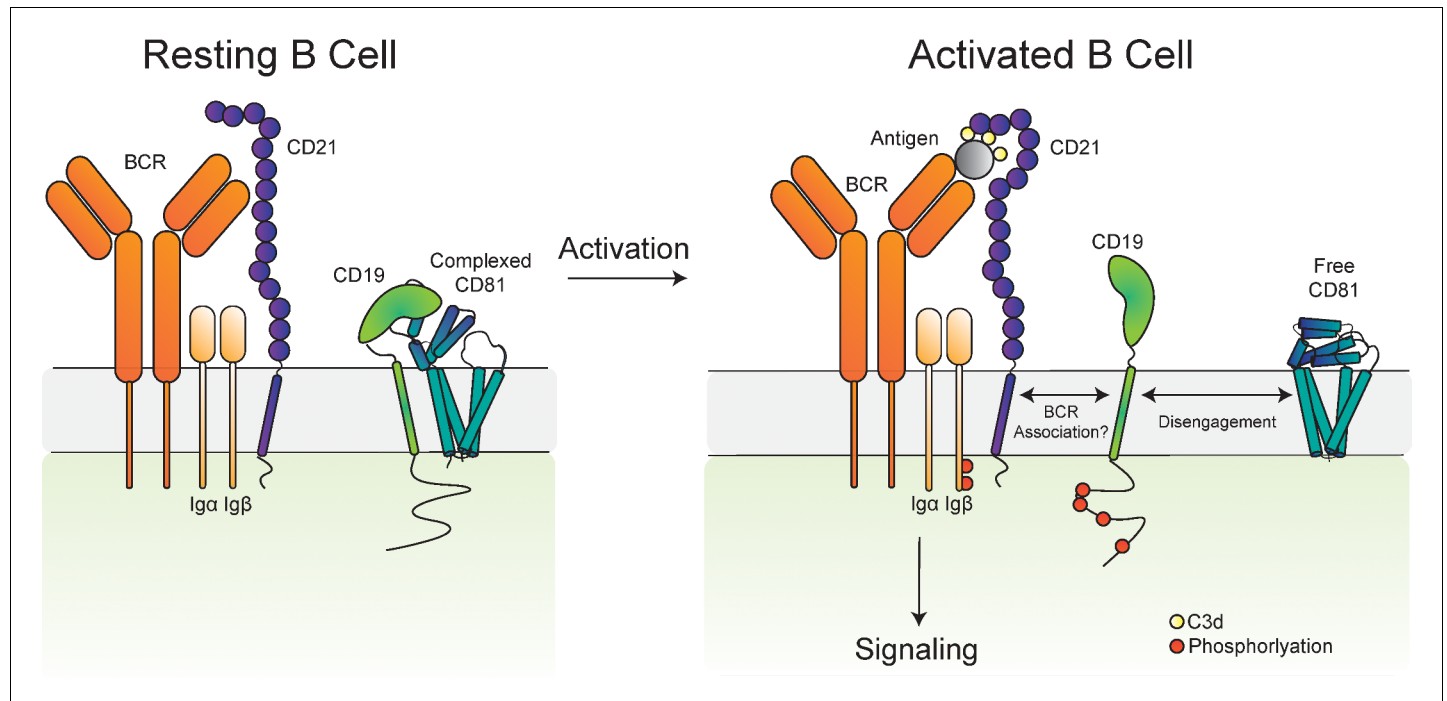

**Figure 5.** Proposed model for the disengagement of the CD81 during B cell activation. Upon B cell activation, dissociation of the B cell co-receptor complex could allow CD19 to freely diffuse in the membrane and interact with the BCR, leading to amplified signaling through the BCR and activation of the B cell.

recently shown promise as a therapeutic lead, since it can selectively target malignant B cells while sparing normal cells (*Vences-Catalán et al., 2019*).

Our observation of regulated dissociation of the CD19-CD81 complex is particularly interesting in view of prior experiments suggesting that CD81 may regulate the diffusion of CD19 by immobilizing it in distinct locations in the membrane (*Mattila et al., 2013*; *Cherukuri et al., 2004*). For example, super resolution microscopy has shown that in resting B cells, the mobility of a large proportion of CD19 is very low, while in CD81-deficient B cells CD19 diffusion shifts strongly to a faster moving population (*Mattila et al., 2013*). This control of CD19 diffusion by CD81 could allow regulation of CD19's interaction with the BCR, for example, to prevent high-level constitutive signaling (*Mattila et al., 2013*). One possible explanation for our data is that, upon B cell activation, dissociation of the CD19-CD81 complex allows CD19 to freely diffuse in the membrane and interact with the BCR, leading to amplified signaling through the BCR and activation of the B cell (*Figure 5*). Consistent with this idea, proximity ligation experiments have also shown that CD19 is in close association with the BCR on activated cells when stimulated through IgM (*Kläsener et al., 2014*). It is also possible that this dissociation is not directly related to the association of CD19 with the BCR, but is instead related to events downstream of BCR engagement. For example, others have shown that CD81 redistributes to the immune synapse of activated B cells (*Mittelbrunn et al., 2002*). Several integrins have been reported to associate with CD81 and are involved in CD81-mediated adhesion in the immune synapse in activated B cells as well as in B cell trafficking to lymphoid organs (*Levy et al., 1998*).

Beyond our finding that the ectodomains of CD19 and CD81 interact to control receptor trafficking and localization, there are other lines of evidence suggesting that the use of the large extracellular loop to bind partner proteins will be a general property of the tetraspanin protein family. First, the transmembrane regions of tetraspanins are highly conserved, but the large extracellular loop varies among family members in both size and sequence. Sequence analysis reveals that within the large extracellular loop, tetraspanins contain a low-polarity hypervariable region that may be involved in tetraspanin binding partner recognition (*Stipp et al., 2003*). Additionally, it is known that several other tetraspanins and their binding partners, such ADAM10 and the C8 family of tetraspanins and the tetraspanin CD151 and integrin $\alpha_3\beta_1$, rely on their ectodomains for binding of partner proteins (*Noy et al., 2016*; *Yauch et al., 2000*). Like the CD19-CD81 complex, other tetraspanin-partner protein complexes may also be dynamically regulated upon changes in cell state, and conformationally specific antibodies may serve as powerful tools to investigate and control tetraspanin biology.

## Materials and methods

**Key resources table**

| Reagent type (species) or resource | Designation | Source or reference | Identifiers | Additional information |
|---|---|---|---|---|
| Antibody | Rabbit Polyclonal CD19 antibody | Cell Signaling | Cat#3574S; RRID:AB_2275523 | 1:500 (western blot) |
| Antibody | CD19 Mouse Monoclonal Antibody (SJ25-C1), Alexa Fluor 488 | Thermo-Fisher | Cat#MHCD1920; RRID:AB_389313 | 2 µg/mL (flow cytometry) |
| Antibody | GAPDH (D16H11) Rabbit Monoclonal Antibody (HRP Conjugate) | Cell Signaling | Cat#8884S; RRID:AB_11129865 | 1:10,000 (western blot) |
| Antibody | APC Mouse Monoclonal anti-human CD81 | BioLegend | Cat#349510; RRID:AB_2564020 | 2 µg/mL (flow cytometry) |
| Antibody | Human Monoclonal CD81 Clone Ab5 | Recombinant; *Nelson et al., 2018* | | 2 µg/mL (flow cytometry) |

*Continued on next page*

*Continued*

| Reagent type (species) or resource | Designation | Source or reference | Identifiers | Additional information |
|---|---|---|---|---|
| Antibody | Human Monoclonal CD81 Clone Ab10 | Recombinant; *Nelson et al., 2018* | | 2 µg/mL (flow cytometry) |
| Antibody | Human Monoclonal CD81 Clone Ab21 | Recombinant; *Nelson et al., 2018* | | 2 µg/mL (flow cytometry) 1:1000 (western blot) |
| Antibody | Human Monoclonal CD81 Clone 5A6 | Recombinant; WO 2017/218691 A1 | | 2 µg/mL (flow cytometry) 1:100 (western blot) |
| Antibody | Human Monoclonal CD19 (Coltuximab) | Recombinant; Therapeutic Antibody Database | | 2 µg/mL (flow cytometry) |
| Antibody | Human Monoclonal CD19 (Denintuzumab) | Recombinant; Therapeutic Antibody Database | | 2 µg/mL (flow cytometry) |
| Antibody | Human Monoclonal CD19 (Inebiliziumab) | Recombinant; Therapeutic Antibody Database | | 2 µg/mL (flow cytometry) |
| Antibody | APC Mouse Monoclonal CD86 antibody | BioLegend | Cat#374208; RRID:AB_2721449 | 2 µg/mL (flow cytometry) |
| Antibody | APC Mouse Monoclonal CD20 Clone L27 | BD Biosciences | Cat#340941; RRID:AB_1645724 | 1 µg/mL (flow cytometry) |
| Antibody | Donkey anti-Rabbit IgG (H+L) HRP Conjugate | Sigma Aldrich | Cat#GENA934-1ML; RRID:AB_2722659 | 1:5000 (western blot) |
| Antibody | Rabbit Anti-Human IgG H and L HRP Conjugate | Abcam | Cat#ab6759; RRID_:AB_955434 | 1:5000 (western blot) |
| Antibody | F(ab')2-Goat anti-Human IgG, IgM (H+L), Functional Grade | Thermo-Fisher | Cat#16-5099-85 | 20 µg/mL (B cell activation) |
| Biological sample (human) | Leuko-reduction Collar | Brigham and Women's Hospital Crimson Core | | |
| Commercial assay or kit | QuickExtract DNA Extraction Solution | VWR | Cat#76081–766 | |
| Chemical compound | Valproic Acid Sodium Salt | Sigma Aldrich | Cat#P4543-25G | |
| Chemical compound, drug | D-(+)-Glucose solution | Sigma Aldrich | Cat#G8769-100ML | |
| Chemical compound, drug | Magnesium Formate DiHydrate | Hampton Research | Cat#HR2-537 | |
| Chemical compound, drug | StockOptions Sodium Acetate | Hampton Research | Cat#HR2-933-01 | |
| Chemical compound, drug | Polyethylene Glycol Monomethyl Ether 550 | Hampton Research | Cat#HR2-611 | |
| Other | MicroTools | Hampton Research | Cat#HR4-837 | |

*Continued on next page*

*Continued*

| Reagent type (species) or resource | Designation | Source or reference | Identifiers | Additional information |
|---|---|---|---|---|
| Commercial assay or kit | RosetteSep Human B Cell Enrichment Cocktail | STEMCELL Technologies | Cat#15024 | |
| Commercial assay or kit | Lymphoprep density gradient medium | STEMCELL Technologies | Cat#07851 | |
| Chemical compound, drug | Nutridoma-SP | Sigma Aldrich | Cat#11011375001 | |
| Chemical compound, drug | n-Dodecyl-B-D-maltoside (DDM) | Anatrace | Cat#D310 | |
| Chemical compound, drug | Cholesteryl Hemisuccinate | Sigma Aldrich | Cat#C6512 | |
| Chemical compound, drug | Iodoacetamide | Sigma Aldrich | Cat# I1149-5G | |
| Peptide, recombinant protein | Benzonase nuclease | Sigma Aldrich | Cat# E1014-25KU | |
| Commercial assay or kit | In-Fusion HD Cloning Plus | Clontech | Cat# 638911 | |
| Cell line (human) | Expi293F Cells | Thermo-Fisher | Cat#A14527 | |
| Cell line (human) | CD81 null HEK293T cells | This paper | | HEK293T cells with CD81 knocked out using CRISPR/Cas9 |
| Sequence-based reagent | gRNA forward primer for CD81 knockout cell line generation | Integrated DNA Technologies | | CACCGATGCGCTGCGTCTGCGGCG |
| Sequence-based reagent | gRNA reverse primer for CD81 knockout cell line generation | Integrated DNA Technologies | | AAACCGCCGCAGACGCAGCGCATC |
| Recombinant DNA reagent | pcDNA3.1 (+) Mammalian Expression Vector | Thermo-Fisher | Cat#V79020 | |
| Recombinant DNA reagent | pFUSE-hIgG1-Fc2 | InvivoGen | Cat#pfuse-hg1fc2 | |
| Recombinant DNA reagent | pD2610-v5 CMV(v5)-ORF Mamm-ElecD | ATUM | N/A | |
| Recombinant DNA reagent | gBlocks | Integrated DNA Technologies | N/A | |
| Recombinant DNA reagent | pSpCas9(BB)—2A-GFP (PX458) | Addgene | Cat#48138 | |
| Software, algorithm | GraphPad Prism 8.0 | N/A | http://www.graphpad.com/scientific-software/prism/ | |

*Continued on next page*

*Continued*

| Reagent type (species) or resource | Designation | Source or reference | Identifiers | Additional information |
|---|---|---|---|---|
| Software, algorithm | BD Accuri C6 Plus software | BD Accuri C6 Plus | http://www.bdbiosciences.com/en-us/instruments/research-instruments/research-cell-analyzers/accuri-c6-plus | |
| Software, algorithm | SB Grid Consortium | *Morin et al., 2013* | https://sbgrid.org/software/ | |
| Software, algorithm | XDS | *Kabsch, 2010* | https://sbgrid.org/software/ | |
| Software, algorithm | Phenix | *Afonine et al., 2012* | https://sbgrid.org/software/ | |
| Software, algorithm | Coot | *Emsley and Cowtan, 2004* | https://sbgrid.org/software/ | |
| Software, algorithm | PyMOL | *DeLano, 2010* | https://sbgrid.org/software/ | |

## Cell lines

HEK293T cells were obtained from the American Type Culture Collection and were confirmed to be negative for mycoplasma contamination prior to use in experiments. No cell lines on the ICLAC register of commonly misidentified cell lines were used in this work.

## CD81 knockout HEK293T cell line

To generate a CD81$^{-/-}$ HEK293T cell line, the CHOPCHOP guide design server (http://chopchop.cbu.uib.no/) was used to select guide sequences targeting exon 1 of the *CD81* gene. The two complementary DNA strands of the guide sequences (IDT Technologies) were annealed in 10 mM Tris pH 8.0, 50 mM NaCl, 1 mM EDTA and then subcloned into a pSpCas9 WT-2A-GFP vector. The resulting pSpCas9 WT-2A-GFP cDNA was transfected into HEK293T cells using polyethyleneimine. Cells expressing GFP were sorted into 96-well plates by flow cytometry 48 hr after transfection. Clonal populations were allowed to expand for 4 weeks. Genomic DNA was extracted from individual clones, and the CD81 gene was amplified by PCR and sequenced to confirm the presence of targeted mutations. The loss of CD81 expression was confirmed by flow cytometry.

## CD19 export assay

CD81$^{-/-}$ HEK293T cells were seeded at 100,000 cells/well in 24 well plates 12–18 hr prior to transfection. CD81$^{-/-}$ HEK293T cells were transfected using Lipofectamine 2000 with either with either 1.5 µg of empty pcDNA3.1(+) vector, 0.75 µg of CD19 DNA and 0.75 µg of empty pcDNA3.1(+) vector DNA (CD19 condition), 0.75 µg of CD19 DNA and 0.75 µg of CD81 DNA (CD19+CD81 condition), or 0.75 µg of CD19 DNA and 0.75 µg of a CD81 chimera DNA. 36–48 hr after transfection, cells were harvested in phosphate buffered saline (PBS) supplemented with 3 mM EDTA, transferred to a 96 well V-bottom plate, and then washed twice with PBS. Cells were then incubated on ice for 20 min with 2 µg/mL Alexa 488-anti-CD19 (ThermoFisher) and APC-anti-CD81 (BioLegend) in 20 mM HEPES buffer pH 7.4, containing 150 mM NaCl, and 0.1% BSA. Cells were washed two times with PBS and analyzed on a BD Accuri C6 flow cytometer.

## Cloning of constructs

### CD19-CD81 fusion protein

The CD19-CD81 fusion was cloned into pcDNA3.1(+) with an N-terminal haemagglutinin signal sequence followed by a FLAG epitope tag and a 3C protease cleavage site. Residues 20–329 of CD19 (ectodomain, transmembrane domain, and first 15 cytoplasmic amino acids) were connected to full length CD81 using a GGSG linker.

## CD81 chimeras

CD81 chimeras were constructed by PCR and subcloned into pcDNA3.1(+). All chimeras were created within the backbones of wild-type human CD9, *C. elegans* Tspan15, or human claudin-4. The following domain boundaries were used:

| Domain | Residue boundaries |
|---|---|
| Large Extracellular Loop CD81 | 117–199 |
| Small Extracellular Loop CD81 | 37–54 |
| First Transmembrane Domain CD81 | 13–33 |
| Helix C of Large Extracellular Loop CD81 | 161–170 |
| Helix D of Large Extracellular Loop CD81 | 181–186 |
| First Transmembrane Domain of Tspan15 *C. elegans* | 21–41 |
| Large Extracellular Loop of Tspan15 *C. elegans* | 115–223 |
| Small Extracellular Loop of Tspan15 *C. elegans* | 42–62 |
| Small Extracellular Loop of CD9 | 34–55 |
| Large Extracellular Loop of CD9 | 112–195 |
| First Transmembrane Domain of CD9 | 13–33 |
| Transmembrane Domain of CD19 | 292–313 |
| Transmembrane Domain of the Sigma One Receptor | 6–32 |

### Antibodies 5A6, Ab5, Ab10, Ab21, Denintuzumab, Coltuximab, and Inebilizumab

The variable regions of each antibody heavy chain were subcloned into the pFUSE-hIgG1-Fc2 vector (Invitrogen). The variable region of the light chains and the human kappa constant sequence with an N terminal 'MDWTWRILFLVAAATGAHS' signal sequence were cloned in the pD2610-v5 vector (ATUM). An additional construct of the 5A6 antibody was also cloned, with a 3C protease site flanked by a Gly-Gly-Ser-Gly linker inserted into the hinge region of the heavy chain, allowing for generation of the 5A6 Fab after cleavage with 3C protease for use in crystallography.

### Validation of CD19-CD81 fusion in CD19 export assay

CD81$^{-/-}$ HEK293T cells were seeded at 100,000 cells/well in 24 well plates 12–18 hr prior to transfection. CD81$^{-/-}$ HEK293T cells were transfected using Lipofectamine 2000 with either 0.75 μg of CD19 DNA and 0.75 μg of empty vector DNA (CD19 condition), 0.75 μg of CD19 DNA and 0.75 μg of CD81 DNA (CD19+CD81 condition), or 0.75 μg of CD19-CD81 DNA and 0.75 μg empty vector DNA (CD19-CD81 fusion protein condition). 48 hr after transfection, cells were harvested in PBS containing 3 mM EDTA, transferred to a 96 well V-bottom plate, and then washed two times with PBS. Cells were then incubated on ice for 20 min with 2 μg/mL Alexa 488-anti-CD19 (ThermoFisher), Coltuximab (recombinant), Denintuzumab (recombinant), or Inebilizumab (recombinant) in 20 mM HEPES buffer pH 7.4, 150 mM NaCl, and 0.1% BSA. Cells stained with recombinant CD19 antibodies were detected with Goat anti Human IgG (H+L) Secondary Antibody, Alexa Fluor 488 (Invitrogen) at 1 μg/mL. Cells were washed two times with PBS and analyzed on a BD Accuri C6 flow cytometer.

### Expression and purification of 5A6 Fab-CD81 LEL complex

The heavy and light chains of the 5A6 antibody and the CD81 large extracellular loop were co-expressed in Expi293F cells. 600 mL of Expi293F cells maintained in Expi293 expression medium were grown to a density of $2.8 \times 10^6$ cells/mL and then transiently transfected with heavy chain 5A6, light chain 5A6, and CD81 LEL DNA (0.48 mg total DNA) and FectoPro transfection reagent (Polyplus) at a 1:1 DNA/FectoPro ratio. 20 hr after transfection, the cells were fed 5 mM Valproic acid sodium salt (Sigma-Aldrich) and 5.5 mL of 45% D-(+)-Glucose solution (Sigma-Aldrich). Transfected cells were cultured for 7 days to produce protein and then the medium was collected and separated

from the cells by centrifugation at 4000 g for 15 min at 4°C. The cultured medium was loaded onto protein A resin (Millipore). The resin was washed with 100 mL 20 mM Tris buffer pH 7.4, containing 150 mM NaCl and then bound protein was eluted in 20 mL 100 mM glycine buffer pH 3.0. Elution fractions were immediately neutralized with 1M Tris buffer pH 7.5. Eluted protein was buffer exchanged into 20 mM Tris buffer pH 7.4, containing 150 mM NaCl and then 3C protease was added at a 1:1 w/w ratio and incubated overnight at 4°C. The purity of the eluted protein and efficiency of cleavage was assessed on an SDS-PAGE Coomassie-stained gel.

The cleaved, recovered 5A6 Fab and CD81 LEL was then applied to nickel resin to select for Fab bound to CD81. Nickel resin was washed with 20 mM Tris buffer pH 7.4, containing 150 mM NaCl and 20 mM imidazole pH 7.4 and then eluted in the same buffer with 350 mM imidazole. The sample was then applied to Protein A resin to remove any residual free Fc. Flow through from the protein A resin was then concentrated with a centrifugal filter, and the purified 5A6 Fab-LEL complex was isolated on an S200 size exclusion column in 20 mM Tris buffer pH 7.4, containing 150 mM NaCl. The purity of fractions corresponding to the 5A6 Fab-LEL complex peak were assessed on an SDS-PAGE Coomassie-stained gel and then pooled and concentrated to 7.2 mg/mL for crystallography. The concentrated complex was stored at 4°C for ~36 hr prior to setting trays.

## Crystallization of 5A6 Fab-CD81 LEL complex

Crystals of the 5A6 Fab-CD81 LEL complex were grown in 96-well sitting drops at room temperature. Branched crystals of the complex (7.5 mg/mL) grew after 48 hr in 0.2 M magnesium formate dihydrate, 0.1 M sodium acetate trihydrate pH 4.0, 18% w/v polyethylene glycol monomethyl ether 5000. Crystals harvested for data collection were grown in 100 nL drops (50 nL protein + 50 nL precipitant) with a 50 µL reservoir solution. To isolate single crystal fragments, crystals were cut with MicroTools (Hampton Research). Crystals were cryoprotected by supplementing the mother liquor with 20% glycerol (v/v). Individual crystals were flash frozen in liquid nitrogen and stored until data collection.

Data collection was performed at Advanced Photon Source NE-CAT beamline 24 ID-C. Diffraction images were processed and scaled using XDS (*Kabsch, 2010*). The crystals were indexed in the space group $P2_12_12_1$ with unit-cell dimensions of 40.0, 96.9, 297.1 Å. Data to a maximum resolution of 2.4 A° were used for structure solution and refinement. The structure was solved by molecular replacement using the program Phaser (*McCoy et al., 2007*). Search models included the heavy and light chain (chain H and L) from PDB entry 4S1D and residues 112–201 of chain A (large extracellular loop of CD81) from PDB entry 5TCX. Two copies of the Fab-CD81 complex were modeled in the asymmetric unit. The structural model was built by auto-build and iterative cycles of manual rebuilding and refinement in Coot and PHENIX (*Afonine et al., 2012*).

## Expression and purification of Ab5, Ab10, Ab21, and 5A6 for use in Flow Cytometry

Expi293F cells maintained in Expi293 expression medium were grown to cell density of $2.8 \times 10^6$ cells/mL and then transiently transfected. For each antibody, plasmids encoding the heavy and light chain were transfected at a 1:2 molar ratio (0.8 mg total DNA/liter cells) with FectoPro transfection reagent (Polyplus) at 1:1 DNA/FectoPro ratio. 20–24 hr after transfection, the cells were fed with 5 mM Valproic acid sodium salt (Sigma-Aldrich) and 45% D-(+)-Glucose solution (Sigma-Aldrich). 4–7 days after transfection, the medium was collected and separated from the cells by centrifugation at 4000 g for 15 min at 4°C. Cultured medium was loaded onto protein A resin (Millipore). The resin was then washed with 50 mL of 20 mM HEPES buffer pH 7.4, containing 150 mM NaCl and then bound protein was eluted in 10 mL of 100 mM glycine buffer, pH 3.0. Elution fractions were immediately neutralized with 1 M HEPES buffer pH 7.4. Eluted protein was buffer exchanged into 20 mM HEPES buffer pH 7.4, containing 150 mM NaCl.

## Isolation of primary human B cells

A leuko-reduction collar was obtained from the Brigham and Women's Hospital Crimson Core with patient information deidentified. All methods were carried out in accordance with relevant guidelines and regulations. All experimental protocols were reviewed and approved as exempt by the Harvard Faculty of Medicine Institutional Review Board. Primary human B cells were isolated from

fresh leuko-reduction collar blood by using 750 µL of RosetteSep Human B Cell Enrichment Cocktail (Stemcell Technologies) for 10 mL of collar blood. The RosetteSep cocktail was incubated with collar blood for 20 min at room temperature, and then 10 mL of PBS supplemented with 2% FBS was added to the collar blood and mixed gently. The diluted collar blood was then layered on top of 10 mL Lymphoprep density gradient medium (Stemcell Technologies). After centrifugation at 1200 g for 20 min, the mononuclear cell layer was harvested and washed twice with PBS supplemented with 2% FBS. Purified cells were then resuspended in warm BD Quantum Yield medium (BD Biosciences) supplemented with 10% FBS and 1:50 Nutridoma-SP (Sigma Aldrich) at $10^6$ cells/mL.

### B cell activation assay

Purified cells were resuspended in warm BD Quantum Yield medium supplemented with 10% FBS and 1:50 Nutridoma-SP (Sigma Aldrich) at $10^6$ cells/mL and allowed to rest for 30 min at 37°C and 5% $CO_2$. Cells were then pipeted several times to disperse clumps and were split into 96 wells for stimulation assays (100 µL per well). Cells were allowed to rest for 1 hr at 37°C and 5% $CO_2$. Either 100 µL of medium (unstimulated condition) or 100 µL of medium containing 20 µg/mL F(ab')2-Goat anti-Human IgG, IgM (H+L) (Invitrogen) was added to each well. 72 hr later, cells were harvested at 500 g for 5 min, washed once with PBS, stained with 2 µg/mL CD69, CD86, CD81, or CD19 antibody in 20 mM HEPES buffer (pH 7.4), 150 mM NaCl, and 0.1% BSA for 20 min on ice, washed twice with PBS, and analyzed on a BD Accuri C6 flow cytometer.

### Western blot analysis of total CD19 and CD81 protein abundance in resting and activated primary human B cells

Cell lysates were run on a non-reducing SDS-PAGE gel, and the protein was transferred to a nitrocellulose membrane. The membrane was then gently rocked in in TBST (0.1% Tween-20 in Tris-buffered-saline) blocking buffer with 5% (w/v) non-fat milk powder at room temperature for 1.5 hr. The blocked membrane was cut and then incubated in TBST with 1% non-fat milk powder containing either GADPH-HRP conjugate antibody (Cell Signaling; 1:10,000 dilution), CD19 SJ25 antibody (Cell Signaling; 1:500 dilution) or CD81 5A6 antibody (Recombinant, 0.4 mg/mL; 1:100 dilution). Antibody incubations were performed overnight at 4°C with shaking. The CD19 blot was incubated with Donkey anti Rabbit IgG (H+L) HRP Conjugate (Thermo Fischer) diluted 1:5000 in TBST containing 1% non-fat milk powder. The CD81 blot was incubated with Rabbit Anti-Human IgG H and L HRP Conjugate (Abcam) diluted 1:5000 in TBST with 1% non-fat milk powder. Secondary antibody incubations were performed for 1 hr at room temperature with shaking. Prior to chemiluminescent detection, blots were washed with TBST three times for 10 min each time. Western blots were developed with Western Lightning Plus-ECL, Enhanced Chemiluminescence Detection Kit (PerkinElmer). Densitometry was performed in ImageJ. CD19 and CD81 levels were normalized to GAPDH.

### CD19-CD81 pulldown and western blots in resting and activated primary human B cells

Purified cells were resuspended in warm BD Quantum Yield medium supplemented with 10% FBS and 1:50 Nutridoma-SP (Sigma Aldrich) at 1 million cells/mL and allowed to rest for 30 min at 37°C and 5% $CO_2$. Cells were then pipeted several times to disperse clumps and were split into a 6 well plate for stimulation (1 mL per well). Cells were allowed to rest for 1 hr at 37°C and 5% $CO_2$. Either 1 mL of medium (unstimulated condition) or 1 mL of medium containing 10 µg/mL F(ab')2-Goat anti-Human IgG, IgM (H+L) (Invitrogen) was added to each well. 72 hr later, cells were harvested at 500 g for 10 min. Cells were lysed in 50 µL 20 mM HEPES buffer pH 7.4, containing 2 mM $MgCl_2$, 2 mg/mL iodoacetamide and 1:100,000 v:v benzonase nuclease. Lysate was centrifuged at 16,000 g for 15 min, and then membranes were resuspended in 200 µL 20 mM HEPES buffer pH 7.4, containing 1% n-Dodecyl-B-D-maltoside (DDM) (Anatrace), 0.1% cholesteryl hemisuccinate (Sigma Aldrich), 250 mM NaCl, and 10% v/v glycerol and then incubated at 4°C with rotating for 2 hr. 200 µL of solubilized protein was then applied to 10 µL protein A resin preincubated with 15 µg Coltuximab (anti-CD19, recombinant) and incubated at 4°C with rotating for 2 hr. Samples were centrifuged at 200 g for 1 min and washed twice with 1 mL of 20 mM HEPES buffer pH 7.4, containing 0.1% n-Dodecyl-B-D-maltoside (DDM) (Anatrace), 0.01% cholesteryl hemisuccinate (Sigma Aldrich), 250 mM NaCl, and 1% v/v glycerol. Samples were eluted in 50 µL 2X SDS loading dye and separated by SDS-PAGE

under non-reducing conditions. The membrane was blocked with 5% (w/v) non-fat milk powder in TBST (0.1% Tween-20 in Tris-buffered-saline) at room temperature for 1.5 hr. The blocked membrane was cut and then incubated with shaking overnight at 4°C with either CD19 SJ25 antibody (Cell Signaling; 1:1000 dilution) or CD81 Ab21 antibody (Recombinant, 1 mg/mL; 1:1000 dilution) in TBST containing 1% non-fat milk powder. The CD19 blot was incubated with donkey anti Rabbit IgG (H+L) HRP Conjugate (Thermo Fischer) diluted 1:5000 in TBST containing 1% non-fat milk powder. The CD81 blot was incubated with Rabbit Anti-Human IgG H and L HRP Conjugate (Abcam) diluted 1:5000 in TBST with 1% non-fat milk powder. Secondary antibody incubations were performed for 1 hr at room temperature with shaking. Prior to chemiluminescent detection, blots were washed with TBST three times for 10 min each time. Western blots were developed with Western Lightning Plus-ECL, Enhanced Chemiluminescence Detection Kit (PerkinElmer). Densitometry was performed in ImageJ. CD81 levels were normalized to CD19.

## Acknowledgements

We would like to thank Izabela Durzynska for cloning of CD9/CD81 chimeras, the CD19 transmembrane domain chimera, and cloning of the guide RNA. We would also like to thank Sanchez Jarrett for assistance with harvesting crystals and Megan Sjodt-Gable for input on figure design. Financial support for this work was provided by NIH grants R35 CA220340 (SCB), F31 HL147459 (KJS) and DP5 OD02134 (ACK).

## Additional information

### Competing interests

Stephen C Blacklow: receives funding for an unrelated project from Novartis, and is a consultant for IFM and Ayala Pharmaceuticals for unrelated projects. Andrew C Kruse: is a consultant on unrelated projects for the Institute for Protein Innovation, a non-profit research institute. The other authors declare that no competing interests exist.

### Funding

| Funder | Grant reference number | Author |
| --- | --- | --- |
| National Institutes of Health | R35 CA220340 | Stephen C Blacklow |
| National Institutes of Health | F31 HL147459 | Katherine J Susa |
| National Institutes of Health | DP5 OD02134 | Andrew C Kruse |

The funders had no role in study design, data collection and interpretation, or the decision to submit the work for publication.

### Author contributions

Katherine J Susa, Conceptualization, Investigation, Methodology; Tom CM Seegar, Investigation; Stephen C Blacklow, Andrew C Kruse, Conceptualization, Supervision, Project administration

### Author ORCIDs

Katherine J Susa ⓘ https://orcid.org/0000-0003-0077-667X
Stephen C Blacklow ⓘ https://orcid.org/0000-0002-6904-1981
Andrew C Kruse ⓘ https://orcid.org/0000-0002-1467-1222

### Ethics

Human subjects: A leuko-reduction collar was obtained from the Brigham and Women's Hospital Crimson Core with patient information de-identified. All methods were carried out in accordance with relevant guidelines and regulations. All experimental protocols were reviewed and approved as exempt by the Harvard Faculty of Medicine Institutional Review Board.

Decision letter and Author response
Decision letter https://doi.org/10.7554/eLife.52337.sa1
Author response https://doi.org/10.7554/eLife.52337.sa2

## Additional files

### Supplementary files

• Transparent reporting form

### Data availability

Diffraction data and refined coordinates have been deposited in the Protein Data Bank under accession code 6U9S.

The following dataset was generated:

| Author(s) | Year | Dataset title | Dataset URL | Database and Identifier |
|---|---|---|---|---|
| Susa KJ, Seegar TCM, Blacklow SC, Kruse AC | 2020 | Crystal structure of human CD81 large extracellular loop in complex with 5A6 Fab | https://www.rcsb.org/structure/6U9S | RCSB Protein Data Bank, 6U9S |

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
