## [Decision Letter]

**Acceptance summary:**

It is now clear that there is data in the literature supportive of the association of CD19 with BCR upon activation. Furthermore, the added discussion on why the chimeras presented in this paper differ from the previous data by Shoham et al. also improve the paper.

**Decision letter after peer review:**

Thank you for submitting your article "A dynamic interaction between CD19 and the tetraspanin CD81 controls B cell co-receptor trafficking" for consideration by *eLife*. Your article has been reviewed by two peer reviewers, one of whom is a member of our Board of Reviewing Editors, and the evaluation has been overseen by Tadatsugu Taniguchi as the Senior Editor. The following individual involved in review of your submission has agreed to reveal their identity: Mark Wright (Reviewer #2).

The reviewers have discussed the reviews with one another and the Reviewing Editor has drafted this decision to help you prepare a revised submission.

Summary:

This is a very well written and interesting manuscript examining the interactions between the tetraspanin protein CD81 and BCR co-receptor CD19. The authors use various chimeric molecules, fusion proteins and CD81-specific antibodies to dissect the interactions between CD19 and CD81. It has long been appreciated that a molecular partner of CD19 – the tetraspanin Cd81- plays a non-redundant role in CD19 biology, however the molecular details of this interaction were unknown. The data adds a strong molecular analysis of this interaction and complement beautifully previous work on the membrane organization of CD19 by CD81 (by Cherukri et al. and Mattila et al). Using convincing data from a combination of immunological and structural techniques that analyze a complex-dependent monoclonal antibody, the authors map the sub-domains of CD81 responsible for the interaction with CD19, and show compelling evidence that this key molecular interaction occurs only in resting B cells. In activated cells, CD81 "releases" CD19.

However, whilst both reviewers were supportive of this paper, there is an obvious "hole". Examining the model shown in Figure 5, the data to support the first half of the model is very convincing. The paper very clearly shows that CD19/CD81 complex does indeed dissociate upon B cell activation. But what about the second half of the model, that the release of CD81 allows the interaction of CD19 with the BCR upon B cell activation? The paper would be significantly strengthened if the data shown in Figure 4E was reinforced with co-precipitation analyses of a BCR/CD19 complex in resting and activated cells, and or a proximity ligation assay/super-resolution analysis to show that the proximity of CD19 to BCR is increased in the activated B cell membrane.

Essential revisions:

1) As described above, conclusive data supporting the second half of their model needs to be added prior to publication. The paper would be significantly strengthened if the data shown in Figure 4E was reinforced with co-precipitation analyses of a BCR/CD19 complex in resting and activated cells, and or a proximity ligation assay/super-resolution analysis to show that the proximity of CD19 to BCR is increased in the activated B cell membrane after release of CD81.

2) In the same figure, there are discrepancies in their interpretation of the antibody data, which need to be addressed. To test how CD19 and CD81 interact in resting and activated conditions, they isolated primary human B cells and activated them through their BCR. While the authors state in the last paragraph of the subsection “The CD19-CD81 complex dissociates in activated B cells”, that there is no difference in the levels of CD81 on the cell surface using Ab21, they state that there is a biologically relevant difference in CD81 staining with Ab 5A6. This is difficult to understand since the difference in MFI on average is larger with Ab21 in the resting versus activated states than Ab 5A6, but the graphs are shown on different scales. The lack of significance with Ab21 is due to more variation of the CD81 staining on the resting cells with Ab21, raising questions of whether the cells could be rested for a longer period of time prior to staining or naive IgD positive B cells could be gated on to normalize the variation in CD81 staining in the resting state? Also, while the authors state that there are no differences in total CD81 levels in Western blots in Figure 4D, this should be quantified across multiple experiments as there seems to be subtle differences that are not discussed. The IP experiment conclusively shows that CD81 is not pulled down with an IP against CD19 in activated cells, yet this should also be quantified.

---

## [Author Response]

Essential revisions:1) As described above, conclusive data supporting the second half of their model needs to be added prior to publication. The paper would be significantly strengthened if the data shown in Figure 4E was reinforced with co-precipitation analyses of a BCR/CD19 complex in resting and activated cells, and or a proximity ligation assay/super-resolution analysis to show that the proximity of CD19 to BCR is increased in the activated B cell membrane after release of CD81.

We proposed the model in Figure 5 by integrating our results with prior information from the literature, and intend for the model not to be conclusive, but rather to be speculative about why the CD19-CD81 complex is dissociating in activated B cells. The reviewer requested that we include a proximity ligation assay or super-resolution analysis to show that the proximity of CD19 to BCR is increased in activated B cells. Klasner et al., 2014, previously used a proximity ligation assay to monitor the interaction of the BCR with CD19 upon B cell activation. Their work showed that CD19 is in close association with the IgM BCR on activated B cells, but not on resting B cells, and in our experiments, we activated the B cells with an anti-IgM IgG Fab’_2_ fragment. Likewise, super-resolution microscopy experiments by Mattila et al., 2013 (DOI: 10.1016/j.immuni.2012.11.019) also detected a local convergence of IgM and CD19, further supporting the interaction of the BCR with CD19 upon B cell activation.

We agree that determining why CD19 dissociates from CD81 is very interesting, and answering this question is an important long-term goal of our studies. To explore this issue, we designed experiments using ascorbate peroxidase 2 (APEX2)-mediated proximity labeling to track the interactions of CD19 with the BCR, CD21, and CD81 on resting B cells and after activation. We used CRISPR/Cas9 to knock-in APEX2 at the Cterminus of CD19 in a B cell line. We are able to observe labeling of CD19-proximal proteins in resting B cells, but we cannot detect labeled proteins 72 h after activation (possibly because of endogenous peroxide quenching), which is the time point we analyzed that shows dissociation of the CD81-CD19 complex. Due to the technical challenges of this experiment and given the prior published work cited above, we believe further experiments along these lines fall outside the scope of this paper.

We have updated Figure 5 to highlight that there are still unknown mechanisms involved in the association of the co-receptor complex with the BCR, and we have updated the text describing our model in the Discussion to address these considerations.

2) In the same figure, there are discrepancies in their interpretation of the antibody data, which need to be addressed. To test how CD19 and CD81 interact in resting and activated conditions, they isolated primary human B cells and activated them through their BCR. While the authors state in the last paragraph of the subsection “The CD19-CD81 complex dissociates in activated B cells”, that there is no difference in the levels of CD81 on the cell surface using Ab21, they state that there is a biologically relevant difference in CD81 staining with Ab 5A6. This is difficult to understand since the difference in MFI on average is larger with Ab21 in the resting versus activated states than Ab 5A6, but the graphs are shown on different scales. The lack of significance with Ab21 is due to more variation of the CD81 staining on the resting cells with Ab21, raising questions of whether the cells could be rested for a longer period of time prior to staining or naive IgD positive B cells could be gated on to normalize the variation in CD81 staining in the resting state? Also, while the authors state that there are no differences in total CD81 levels in Western blots in Figure 4D, this should be quantified across multiple experiments as there seems to be subtle differences that are not discussed. The IP experiment conclusively shows that CD81 is not pulled down with an IP against CD19 in activated cells, yet this should also be quantified.

The MFI value for bound antibody can be influenced by a variety of factors, including the extent of fluorescein labeling, as well as the accessibility of the bound epitope and the lifetime of the bound state. The differences in absolute MFI observed for Ab21 and 5A6 surface staining is most likely accounted for by a different extent of labeling with fluorescein. Regardless of the origin of the difference in intensity, it is clear that 5A6 surface staining doubles upon activation (with a P value <0.0001), whereas the difference in Ab21 surface staining is a reduction of roughly 10%, which is within the error of the measurement.

As requested, we have quantified the Western blots in Figure 4D and Figure 4E using densitometry. Figure 4 has been updated to include panels F and G with this quantification.